# Study of efficacy and longevity of immune response to third and fourth doses of COVID-19 vaccines in patients with cancer: A single arm clinical trial

Astha Thakkar[1], Kith Pradhan[2], Benjamin Duva[1], Juan Manuel Carreno[3,4], Srabani Sahu[1], Victor Thiruthuvanathan[1], Sean Campbell[5], Sonia Gallego[1], Tushar D Bhagat[1], Johanna Rivera[6], Gaurav Choudhary[1], Raul Olea[1], Maite Sabalza[7], Lauren C Shapiro[1], Matthew Lee[1], Ryann Quinn[1], Ioannis Mantzaris[1], Edward Chu[1], Britta Will[1], Liise-anne Pirofski[6], Florian Krammer[3,4,8], Amit Verma[1]*, Balazs Halmos[1]*

[1]Department of Oncology, Montefiore Einstein Cancer Center, Albert Einstein College of Medicine, Bronx, United States; [2]Department of Epidemiology and Population Health, Albert Einstein College of Medicine, Bronx, United States; [3]Department of Microbiology, Icahn School of Medicine at Mount Sinai, New York, United States; [4]Center for Vaccine Research and Pandemic Preparedness (C-VARPP), Icahn School of Medicine at Mount Sinai, New York, United States; [5]Department of Pathology, Montefiore Medical Center, Bronx, United States; [6]Department of Medicine, Albert Einstein College of Medicine, Bronx, United States; [7]Euroimmun, Mountain Lakes, United States; [8]Department of Pathology, Molecular and Cell-Based Medicine, Icahn School of Medicine at Mount Sinai, New York, United States

*For correspondence:
amit.verma@einsteinmed.edu
(AV);
bahalmos@Montefiore.org (BH)

## Abstract

**Background:** Cancer patients show increased morbidity with COVID-19 and need effective immunization strategies. Many healthcare regulatory agencies recommend administering 'booster' doses of COVID-19 vaccines beyond the standard two-dose series, for this group of patients. Therefore, studying the efficacy of these additional vaccine doses against SARS-CoV-2 and variants of concern is of utmost importance in this immunocompromised patient population

**Methods:** We conducted a prospective single arm clinical trial enrolling patients with cancer that had received two doses of mRNA or one dose of AD26.CoV2.S vaccine and administered a third dose of mRNA vaccine. We further enrolled patients that had no or low responses to three mRNA COVID vaccines and assessed the efficacy of a fourth dose of mRNA vaccine. Efficacy was assessed by changes in anti-spike antibody, T-cell activity, and neutralization activity, which were again assessed at baseline and 4 weeks.

**Results:** We demonstrate that a third dose of COVID-19 vaccine leads to seroconversion in 57% of patients that were seronegative after primary vaccination series. The immune response is durable as assessed by anti-SARS-CoV-2 (anti-S) antibody titers, T-cell activity, and neutralization activity against wild-type (WT) SARS-CoV2 and BA1.1.529 at 6 months of follow-up. A subset of severely

immunocompromised hematologic malignancy patients that were unable to mount an adequate immune response (titer <1000 AU/mL) after the third dose and were treated with a fourth dose in a prospective clinical trial which led to adequate immune boost in 67% of patients. Low baseline IgM levels and CD19 counts were associated with inadequate seroconversion. Booster doses induced limited neutralization activity against the Omicron variant.

**Conclusions:** These results indicate that third dose of COVID vaccine induces durable immunity in cancer patients and an additional dose can further stimulate immunity in a subset of patients with inadequate response.

**Funding:** Leukemia Lymphoma Society, National Cancer Institute.

**Clinical trial number:** NCT05016622.

## Editor's evaluation

This important study evaluates the immunogenicity of 3rd and 4th doses of SARS-CoV2 vaccinations in patients with cancer. Their study is notable in that neutralization of Omicron was absent in all patients after the third dose but increased to 33% after the fourth dose. With the definitions and patient population better described, this paper would be of interest to those studying the effects of repeated COVID boosters on Omicron immunity.

## Introduction

It is now well established that coronavirus disease 2019 (COVID-19) in patients with cancer carries a higher morbidity and mortality, especially in patients with hematologic malignancies (*Kuderer et al., 2020*; *Lee et al., 2020*; *Mehta et al., 2020*; *Khoury et al., 2022*; *Tang and Hu, 2020*). While overall case fatality has decreased over time, mostly related to the impact of broad vaccinations and improved supportive/antimicrobial management, a higher case fatality rate was noted among cancer patients even during the Omicron (B.1.1.529) wave (*Lee et al., 2022*; *Pinato et al., 2022*; *Bestvina et al., 2022*). Advanced age, co-morbidities, and performance status have emerged as key factors adversely impacting outcomes among patients with a cancer diagnosis (*Grivas et al., 2021*). Effective vaccines have been developed and authorized by the FDA to combat this pandemic (*Sadoff et al., 2021*; *Polack et al., 2020*). However, emerging data suggests that despite these vaccines inducing high levels of immunity in the general population, patients with hematologic malignancies have lower rates of seroconversion as defined by severe acute respiratory syndrome coronavirus 2 (SARS-CoV-2) spike antibody (anti-S antibody) titers (*Thakkar et al., 2021*; *Addeo et al., 2021*). Evidence has also suggested that specific therapies, such as anti-CD20 antibodies, BTK-inhibitors, and stem cell transplantation (SCT), have an association with lower rates of seroconversion (*Schönlein et al., 2022*; *Guven et al., 2022*; *Dahiya et al., 2022*).

We previously published preliminary results of a study defining notable impacts of a third dose of vaccine, demonstrating a more than 50% seroconversion rate among patients remaining seronegative after primary vaccination series of two mRNA vaccine or one adenoviral vaccine (*Shapiro et al., 2022*). Since then, we have completed our entire primary cohort to assess initial responses with a broad array of immunological assays along with now additional significant follow-up allowing assessment of key aspects of waning immunity. Importantly, we additionally conducted a trial assessing the efficacy of a fourth dose of the COVID-19 vaccine among a highly immune suppressed group of patients with no or limited response to three-vaccine doses. Here, we present results of both key cohorts including results of serological, T-cell, and neutralization assays as well as correlations with other baseline clinical, treatment, and laboratory parameters.

**eLife digest** People with cancer have a higher risk of death or severe complications from COVID-19. As a result, vaccinating cancer patients against COVID-19 is critical. But patients with cancer, particularly blood or lymphatic system cancers, are less likely to develop protective immunity after COVID-19 vaccination.

Immune suppression caused by cancer or cancer therapies may explain the poor vaccine response. Booster doses of the vaccine may improve the vaccine response in patients with cancer. But limited information is available about how well booster doses protect patients with cancer against COVID-19.

Thakkar et al. show that a third dose of a COVID-19 vaccine can induce a protective immune response in half of the patients with cancer with no immunity after the first two doses. In the experiments, Thakkar et al. tracked the immune reaction to COVID-19 booster shots in 106 cancer patients. A third booster dose protected patients for up to four to six months and reduced breakthrough infection rates to low levels. Eighteen patients with blood cancers and severe immune suppression had an inadequate immune response after three doses of the vaccine; a fourth dose boosted the immune response for two-thirds of them, which for some included neutralization of variants such as Omicron.

The experiments show that booster doses can increase COVID-19 vaccine protection for patients with cancer, even those who do not respond to the initial vaccine series. Thakkar et al. also show that pre-vaccine levels of two molecules linked to the immune system, (immunoglobin M and the CD19 antigen) predicted the patients' vaccine response, which might help physicians identify which individuals would benefit from booster doses.

## Methods

**Key resources table**

| Reagent type (species) or resource | Designation | Source or reference | Identifiers | Additional information |
|---|---|---|---|---|
| Software, algorithm | RStudio, v3.6.2 | posit | RRID:SCR_000432 | |
| Commercial assay, kit | AdviseDx Abbott SARS-CoV-2 anti-S antibody assay | Abbott | I1000SR instrument | |
| Other | cPass SARS-CoV-2 Neutralization Antibody Detection Kit | GenScript | L00847 | EUA by FDA; https://www.genscript.com/covid-19-detection-cpass.html |
| Other | Quan-T-Cell SARS-CoV-2 and Quan-T-Cell ELISA | EUROIMMUN | ET 2606 and EQ 6841 | CE-marked and for Research Use Only in the United States https://www.coronavirus-diagnostics.com/immune-response-test-systems-for-covid-19.html IFN-γ ELISA: plasma diluted 1:5 |
| Other | mAb 1 C7C7 anti-SARS nucleoprotein antibody | Center for Therapeutic Antibody Development at the ISMMS (Same clone as Sigma Millipore) | ZMS1075 | Working dilution 1 μg/ml |
| Other | (H&L) Antibody Peroxidase ConjugatedGoat Polyclonal | Rockland | 610–1302 | 1:3000 dilution |
| Other | SIGMAFAST OPD (o-Phenylenediamine dihydrochloride) | Sigma-Aldrich | Cat# P9187 | |
| Other | 3-molar hydrochloric acid | Thermo Fisher Scientific | Cat# S25856 | |

### Patient recruitment and follow-up (ClinicalTrials.gov identifier NCT05016622)

Third dose study

We recruited patients via an informed consent process. Patients were required to be >18 years of age and have a cancer diagnosis either on active treatment or requiring active surveillance. Patients were also required to have received two doses of the mRNA COVID-19 vaccine or one dose of the adenoviral

vaccine prior to enrollment. After drawing baseline labs that included spike antibody, a sample for T-cell assay, and a biobank sample, patients received a third mRNA vaccine (initially BNT162b2 per protocol, which was later amended to allow for third mRNA-1273 vaccine after the Food and Drug Administration [FDA] authorized 'booster' doses in fall of 2021). Patients who had received Ad26. CoV2.S vaccine received a BNT162b2 vaccine. The patients then returned for follow-up 4 weeks and 4–6 months after their third dose and their labs were repeated (*Figure 1—figure supplement 1*).

## Fourth dose study

We have previously reported preliminary findings of a 56% seroconversion rate after third dose of vaccine patients with cancer who did not have a detectable immune response after two doses (*Shapiro et al., 2022*). For patients who did not seroconvert after three doses or had low antibody response (<1000 AU/mL as determined by our in-house assay, Abbott), we hypothesized whether a 'mix and match' strategy with fourth dose of COVID-19 vaccine would induce seroconversion/ improved boosting of the humoral antibody responses. To study this, we designed a protocol wherein patients who had received three prior doses of mRNA vaccines and had undetectable anti-S antibody or had an anti-S antibody level of <1000 AU/mL measured at least 14 days after third dose would be randomized to an mRNA vs. adenoviral fourth vaccine dose. Responses would be then assessed at 4 weeks after the fourth dose through measurement of anti-S antibody results. We also measured complete blood counts (CBC), quantitative immunoglobulin levels (IgG, IgA, and IgM), lymphocyte subsets, T-cell responses, and neutralization activity at baseline and 4 weeks for each of these patients. Following the implementation of this protocol, the Centers for Disease Control (CDC) published a statement that advised that the mRNA vaccines should be preferentially administered over the adenoviral vaccines given concern over rare side effects such as thrombocytopenia and thrombosis syndrome. Following this advisory, we amended our protocol to allow recruitment in a cohort that would receive a fourth dose of the BNT162b2 vaccine to comply with CDC guidelines (*Figure 1—figure supplement 1*).

## Anti-S antibody assay

The AdviseDx SARS-CoV-2 IgG II assay was used for the assessment of anti-S IgG antibody. AdviseDx is an automated, two-step chemiluminescent immunoassay performed on the Abbott i1000SR instrument. The assay is designed to detect IgG antibodies directed against the receptor binding domain (RBD) of the S1 subunit of the spike protein of SARS-CoV-2. The RBD is a portion of the S1 subunit of the viral spike protein and has high affinity for the angiotensin converting enzyme 2 (ACE2) receptor on the cellular membrane (*Pillay, 2020*; *Yang et al., 2020*) The procedure, in brief, is as follows. Patient serum containing IgG antibodies directed against the RBD is bound to microparticles coated with SARS-CoV-2 antigen. The mixture is then washed of unbound IgG and anti-human IgG, acridinium-labeled, secondary antibody is added and incubated. Following another wash, sodium hydroxide is added and the acridinium undergoes an oxidative reaction, which releases light energy which is detected by the instrument and expressed as relative light units (RLU). There is a direct relationship between the amount of anti-spike IgG antibody and the RLU detected by the system optics. The RLU values are fit to a logistic curve which was used to calibrate the instrument and expresses results as a concentration in AU/mL (arbitrary units/milliliter) (conversion for spike antibody titers from AU/mL to BAU/mL: based on the results from the first WHO International Standard study, which demonstrated a strong correlation with the current standardization of the SARS-CoV-2 IgG II Quant assays, the mathematical relationship of the Abbott AU/mL unit to WHO unit [binding antibody unit per mL (BAU/mL)] would follow the equation: BAU/mL = 0.142*AU/mL). This assay recently has shown high sensitivity (100%) and positive percent agreement with other platforms including a surrogate neutralization assay (*Bradley et al., 2021*) and also demonstrated high specificity both in the post-SARS-CoV-2 infection and post-vaccination settings. The cutoff value for this assay is 50 AU/mL with <50 AU/mL values reported as negative and the maximum value is 50,000 AU/mL.

## SARS-CoV-2 interferon gamma release assay

The EUROIMMUN SARS-COV-2 interferon gamma release assay (IGRA) (Quan-T-Cell SARS-CoV-2) was used for the assessment of patients' T-cell response to SARS-CoV-2 antigens through analysis of the production of interferon gamma by patient T cells after exposure to SARS-CoV-2-specific proteins.

The assay does not differentiate between vaccine- or infection-induced T-cell responses. The SARS-CoV-2 IGRA is performed in two steps as per the manufacturer's instructions, and a brief protocol follows. First, patient samples from lithium heparin vacutainers are aliquoted into three separate tubes each. These tubes contain either nothing ('blank'), general T-cell activating proteins ('mitogen'), or components of the S1 domain of SARS-CoV-2 ('SARS-CoV-2 activated'). These samples were incubated at 37°C for 24 hr before being centrifuged and the plasma separated and frozen at –80°C for later analysis. Samples were then batched to be run as a full 96-well plate along with calibrators and controls. Plasma samples were unfrozen and added to an ELISA plate, which was prepared with monoclonal interferon-gamma binding antibodies, along with calibrators and controls. After incubation at RT the plate was washed and biotin-labeled anti-interferon gamma antibody was added to bind the patient interferon gamma bound to the plate. The plate was again incubated before being washed of excess antibody and a streptavidin-bound horseradish peroxidase (HRP) enzyme added, which binds strongly to the biotin-labeled antibodies present. This was again incubated and then washed of excess enzyme before a solution of $H_2O_2$ and TMB (3,3',5,5-tetramethylbenzidine, a peroxide-reactive chromogen) is added and allowed to react in the dark for 20 min. The reaction is then stopped through the addition of sulfuric acid and the results read at 450 nM with background subtraction at 650 nM. Results for controls and samples were quantified by the calibration curve generated on the same plate, and results were interpreted as long as controls were within the pre-specified range. Blank results for each specimen set were subtracted from each tube in the set and the mIU/mL for both the mitogen and SARS-CoV-2 activated samples were determined with the calibration curve. Samples with mitogen results below 400 mIU/mL were considered 'invalid', as the overall T-cell activity for that set was too low and excluded from analysis. All other sample sets were interpreted as per the manufacturer's instructions based on the SARS-CoV-2 activated sample results: less than 100 mIU/mL were denoted as negative, and greater than or equal to 100 mIU/mL were denoted as positive.

## Neutralization assays

### Surrogate virus neutralization assay for WT SARS-CoV-2

The SARS-CoV-2 Surrogate Virus Neutralization Test Kit was used to measure antibodies that inhibit the interaction between viral RBD and ACE2 receptor. This test kit uses purified human ACE2 (hACE2) protein-coated enzyme-linked immunosorbent assay (ELISA) plates and HRP-conjugated RBD to monitor the presence of circulating antibodies in samples, including peripheral/capillary blood, serum, and plasma, which block the interaction of RBD-HRP with ACE2 with excellent correlation with the gold standard live virus plaque reduction neutralization test.

The kit contains two key components: RBD-HRP and hACE2. The protein-protein interaction between RBD-HRP and hACE2 is disrupted by neutralizing antibodies against SARS-CoV-2 RBD, if present in a sample. After mixing the sample dilutions with the RBD-HRP solution, components are allowed to bind to the RBD. The neutralization antibody complexed to RBD-HRP remains in the supernatant and is removed during washing, The yellow color of the hACE2-coated wells is determined by the RBD HRP binding to the hACE2-coated wells after incubation with TMB, followed by a stop solution. After the addition of the stop solution, a light-yellow color results from blocking agents interacting with RBDs and inhibiting hACE2 interactions.

### Microneutralization assay

Microneutralization assays were performed in a biosafety level 3 facility at the Icahn School of Medicine at Mount Sinai (ISMMS) as previously described (*Carreño et al., 2022*). Briefly, Vero E6-TMPRSS2 cells were seeded in 96-well cell culture plates at 20,000/well in complete Dulbecco's Modified Eagle Medium (cDMEM). The following day, heat-inactivated serum samples were serially diluted (threefold) starting at a 1:10 dilution in 1× MEM (10× minimal essential medium [Gibco], 2 mM L-glutamine, 0.1% sodium bicarbonate [Gibco], 10 mM 4-(2-hydroxyethyl)-1-piperazineethanesulfonic acid [HEPES; Gibco], 100 U/mL penicillin, 100 μg/mL streptomycin [Gibco], and 0.2% bovine serum albumin [MP Biomedicals]) supplemented with 10% fetal bovine serum (FBS). The virus diluted at 10,000 tissue culture infectious dose 50% ($TCID_{50}$) per mL of 1× MEM was added to the serum dilutions and incubated for 1 hr at room temperature (RT). After removal of cDMEM from Vero E6 cells, 120 μL/well of the virus-serum mix were added to the cells and plates were incubated at 37°C for 1 hr. Mix was removed and 100 μL/well of each corresponding serum dilutions were added in a mirror fashion to

the cell plates. Additional 100 μL/well of 1× MEM 1% FBS (Corning) were added to the cells. Plates were incubated for 48 hr at 37°C and fixed with a 10% paraformaldehyde solution (PFA, Polysciences) for 24 hr at 4°C.

For staining, plates were washed with 200 μL of PBS. Cells were permeabilized with 150 μL/well PBS containing 0.1% Triton X-100 for 15 min at RT. Plates were washed 3× with PBST and blocked with 3% milk (American Bio) in PBST for 1 hr at RT. Blocking solution was removed and 100 μL/well of the biotinylated mAb 1C7C7 anti-SARS nucleoprotein antibody (generated at the Center for Therapeutic Antibody Development at the ISMMS) were added at 2 μg/mL for 1 hr at RT. Plates were then washed 3× with PBST and the secondary antibody goat anti-mouse IgG-HRP (Rockland Immunochemicals) was added at 1:3000 in blocking solution for 1 hr at RT. Plates were washed 3× with PBST, and SIGMAFAST OPD (o-phenylenediamine, Sigma-Aldrich) was added for 10 min at RT. The reaction was stopped with 50 μL/well 3 M hydrochloric acid to the mixture. Optical density ($OD_{490}$) was measured on an automated plater reader (Sinergy 4, BioTek). The inhibitory dilution 50% were calculated as previously described (*Amanat et al., 2020*).

## Statistical analysis

The primary endpoint of the third dose study was to assess the rate of booster-induced seroconversion among patients who remained seronegative at least 28 days following standard set of FDA authorized COVID-19 vaccinations. We hypothesized that booster dosing would convert at least 30% of the enrolled seronegative patients to seropositive as defined by our institutional Clinical Laboratory Improvement Amendments (CLIA) certified SARS-CoV-2 spike IgG assay (as compared to 10% as our null hypothesis). In a pre-specified analysis, at least 26 evaluable seronegative patients were required to have sufficient power to be able to reach this assessment. A McNemar's test was used to determine the equality of marginal frequencies for paired nominal data with the aid of a homogeneity of stratum effects test to check if the effect was the same across all levels of a stratifying variable (*Zhao et al., 2014*). A Wilcox test was used to determine if titers of two paired observations have changed over time subsequently using a Kruskal Wallis test to determine if this difference is associated with another variable. For the fourth dose study, a responder was considered any patient who showed seroconversion from negative anti-S antibody to positive anti-S antibody at 4 weeks after fourth dose or increase in titer to >1000 AU/mL at 4 weeks after the fourth dose. An alpha <0.05 was considered statistically significant. Correlation between continuous variables was assessed using Spearman's test. All analyses were performed in R (version 3.6.2). This study was approved by The Albert Einstein College of Medicine Institutional Review Board.

## Results

### Duration of immune responsiveness after third dose of COVID vaccine in cancer patients

#### Baseline characteristics

We previously reported outcomes for 88 patients enrolled into this study (*Shapiro et al., 2022*). Here, we present our final results for the complete cohort of 106 patients that were enrolled into this study for assessment of the primary endpoint of response at 4 weeks as well as 47 patients who completed 4–6 month follow-up. The baseline characteristics of this cohort are summarized in *Table 1*. The median age was 68 years (63.25–76.5 years). Fifty-five percent (58/106) of patients were female and 45% (48/106) were male. Our cohort was ethnically diverse and included 34% (36/106) Caucasian, 31% (33/106) African-American, 25% (27/106) Hispanic, and 8% (9/106) Asian patients. Majority of patients had received mRNA vaccines at baseline. Sixty-eight percent (72/106) received BNT162b2, 26% (28/106) received mRNA-1273, and 6% (6/106) had received Ad26.CoV2.S. Seventy-four percent of patients (78/106) received a booster BNT162b2 vaccine and 26% (28/106) patients received booster mRNA-1273 vaccine. The majority of the patients, 62% (66/106), had a hematologic malignancy and 38% (40/106) had a solid tumor diagnosis. Further breakdown of cancer type and cancer status is summarized in *Table 1*. The majority of patients, 75% (80/106), were being actively treated at the time of receiving the third dose of the vaccine.

**Table 1.** Baseline characteristics for third dose cohort.

| Baseline characteristics | n=106 |
|---|---|
| Age (median, IQR) | 68 (63.25–76.5) |
| Sex | |
| Male | 48 (45%) |
| Female | 58 (55%) |
| Race | |
| Caucasian | 36 (34%) |
| African-American | 33 (31%) |
| Hispanic | 27 (25%) |
| Asian | 9 (8%) |
| Other | 1 (1%) |
| Previous vaccine given | |
| BNT162b2 | 72 (68%) |
| mRNA-1273 | 28 (26%) |
| Ad26.CoV2.S | 6 (6%) |
| Type of booster vaccine | |
| BNT162b2 | 78 (74%) |
| mRNA-1273 | 28 (26%) |
| Malignancy category | |
| Hematologic malignancy | 66 (62%) |
| Solid Malignancy | 40 (38%) |
| Lymphoid/myeloid/solid | |
| Lymphoid | 55 (52%) |
| Myeloid | 11 (10%) |
| Solid | 40 (38%) |
| Cancer status | |
| Active | 69 (65%) |
| Progressive | 3 (3%) |
| Recurrent | 3 (3%) |
| Relapse | 7 (7%) |
| Remission | 24 (23%) |
| On treatment at the time of booster | |
| Yes | 80 (75%) |
| No | 26 (25%) |

## Serology results

Thirty-three percent of the patients (35/106) were seronegative after two doses. At 4 weeks following the receipt of the booster vaccine, 57% (20/35) of these patients seroconverted and had a detectable antibody response as demonstrated by anti-S antibody testing, meeting the primary endpoint of our study. The median titer at baseline (after primary vaccination) for the entire cohort was 212.1 AU/mL (IQR 50–2873 AU/mL) and the median titer at 4 weeks (after third dose of the vaccine) for the entire cohort was 9997 AU/mL (IQR 880.7–47,063 AU/mL) (*Figure 1A*). The median rise in anti-S titer for

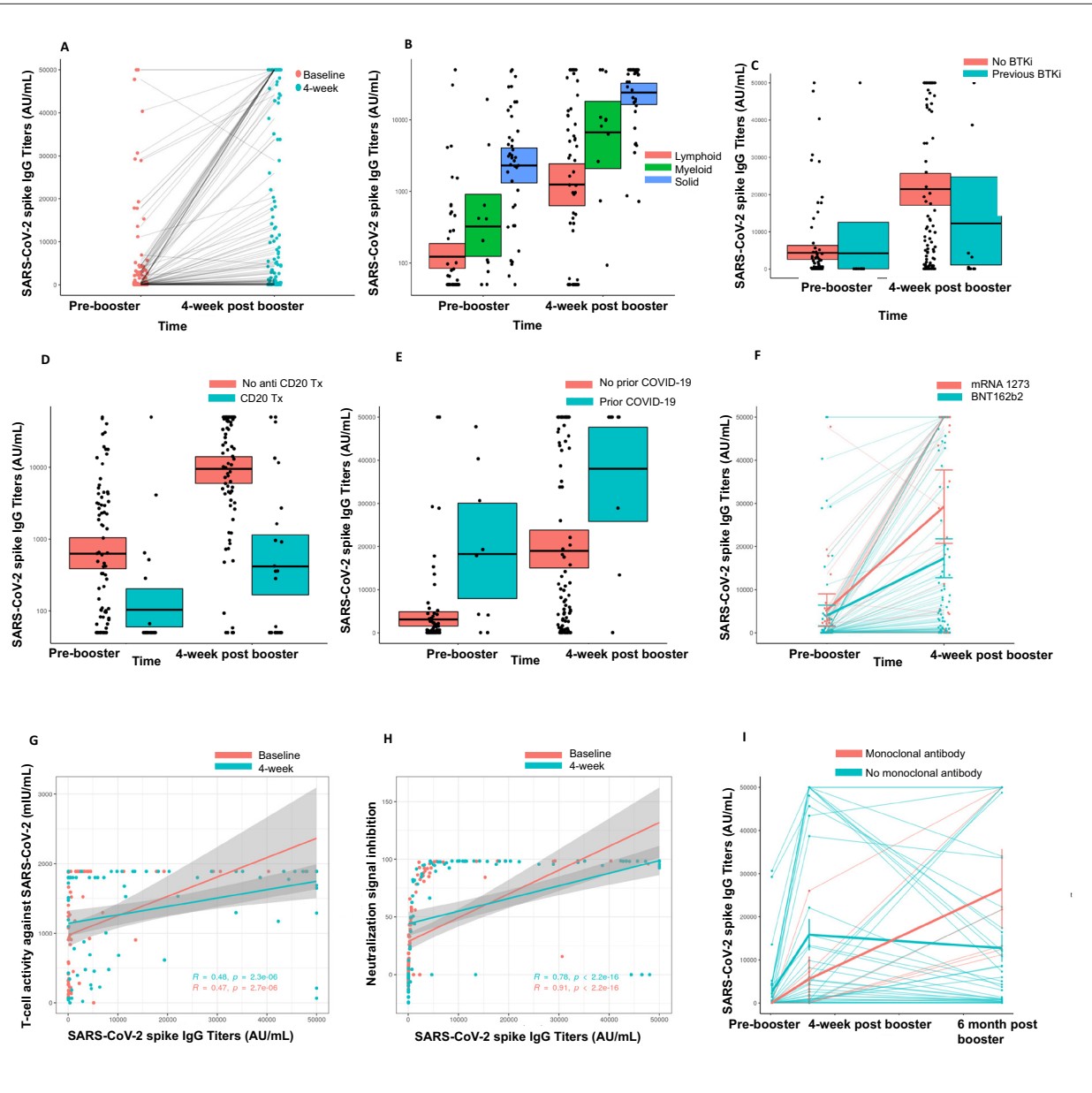

**Figure 1.** Immunogenicity of third dose of coronavirus disease 2019 (COVID-19) vaccine in seronegative cancer patients. (**A**) Figure showing change in anti-SARS-CoV-2 (anti-S) antibody titer at 4 weeks for entire cohort n=106. (**B**) Figure showing change in anti-S antibody titer at 4 weeks split by cancer type (solid cancer, lymphoid cancer, and myeloid cancer) n=106. (**C**) Figure showing effect of Bruton's tyrosine kinase inhibitor (BTKi) therapy on anti-S antibody titer at baseline and 4 weeks of third dose n=12 patients that received BTKi Kruskal-Wallis test. (**D**) Figure showing effect of anti-CD20 antibody therapy on anti-S antibody titer at baseline and 4 weeks of third dose n=25 patients that received anti-CD20 antibody, Kruskal-Wallis test. (**E**) Figure showing effect of prior COVID-19 infection on anti-S antibody titer at baseline and 4 weeks of third dose n=9 patients with COVID infection, Kruskal-Wallis test. (**F**) Figure showing effect of booster type (BNT162b2 vs mRNA 1273) on anti-S antibody titer at baseline and 4 weeks of third dose. (**G**) Line diagram showing correlation between anti-spike IgG titer and baseline T-cell activity at baseline and 4 weeks n=88 for baseline, n=89 for 4 weeks; Spearman's test. (**H**) Line diagram showing correlation between anti-S titer and signal inhibition for neutralization against wild-type (WT) virus at baseline and 4 weeks. n=103 for baseline, n=100 for 4 weeks; Spearman's test. (**I**) Anti-spike IgG titers at baseline, 4 weeks, and 6 months after third dose of COVID-19 vaccine in cancer patients. Line shows means with error bars (SD).n=47. All statistical tests performed at a pre-determined threshold of p<0.05 for statistical significance.

The online version of this article includes the following figure supplement(s) for figure 1:

**Figure supplement 1.** CONSORT diagram showing enrollment and follow-up of study subjects.

patients with hematologic malignancies was 2167 AU/mL (IQR 0–10,131 AU/mL) versus 31,010 AU/mL (IQR 9531–44,464 AU/mL) in patients with solid malignancies (p<0.001). Within the hematologic malignancies, patients with lymphoid cancers had a lower rise in median anti-S titers (1169 AU/mL, IQR 0–8661 AU/mL) compared to those with myeloid malignancies; median anti-S titer 9424 AU/mL (IQR 4381–20,444 AU/mL) (p<0.001) (*Figure 1B*).

We further investigated the association of specific anti-cancer therapies with the booster effect. Patients on Bruton's tyrosine kinase inhibitor (BTKi) therapy (n=12) had a median rise in anti-S antibody of 0 AU/mL (IQR 0–3393 AU/mL) compared to a median rise of 9355 AU/mL (IQR 877.3–34410 AU/mL) in anti-S antibody for patients not on BTKi (p<0.05) (*Figure 1C*). Patients on anti-CD20 antibody therapy (n=25) also had a median rise in anti-S antibody level of 0 AU/mL (IQR 0–910.5 AU/mL) compared to a median rise of 12,735 AU/mL (IQR 2842–38,863 AU/mL) in patients that did not receive anti-CD20 antibody therapy (p<0.05). (*Figure 1D*). Nine patients had a history of SARS-CoV-2 infection and in this cohort the rise in anti-S titers was higher (median 19,350 AU/mL, IQR 9286–32,151 AU/mL), compared to those who did not have prior SARS-CoV-2 infection with a median anti-S titer rise, 6706 AU/mL (IQR 444.1–33,831 AU/mL) (*Figure 1E*). We also observed that the rise in anti-S titer at 4 weeks was higher for patients who received an mRNA-1273 booster compared to BNT162b2 booster; median 31,451 AU/mL vs. 5534 AU/mL, respectively (*Figure 1F*). This observation was not, however, statistically significant. Lastly, we also investigated the association of age with spike antibody response at 4 weeks. The median spike antibody titer for patients <65 years of age was 27,451 AU/mL and the median patients with age ≥65 years was 6152 AU/mL. This result was significant at p value 0.03438. These results are also summarized in *Table 2*.

## T-cell immune responses

We also studied T-cell immune responses through a SARS-CoV-2 IGRA. At baseline (i.e. after primary vaccination), 88 patients had evaluable T-cell results and a positive T-cell response against SARS-CoV-2 was seen in 74% (65/88) patients. Of these 65 patients, 21 patients were seronegative for anti-S antibody at baseline. At 4 weeks (after third dose), 89 patients had evaluable T-cell results and a positive result was seen in 85% (76/89) patients. Of the 15 patients with negative anti-S antibody at 4 weeks, 11 had a positive T-cell response. Fourteen patients who had a negative T-cell assay response at baseline had a positive T-cell response at 4 weeks. Anti-S titer showed a positive correlation with T-cell response at baseline and at 4 weeks for this cohort (p<0.001) (*Figure 1G*). These results are summarized in *Table 2*.

## Neutralization assays

### Neutralization assay against WT virus

We tested neutralization pre- and post-third dose in this cohort using the GenScript surrogate virus neutralization assay. At baseline, biobanked samples from 103 patients were tested for neutralizing antibodies. Of these, 35 patients were seronegative at baseline and 68 patients were seropositive. Neutralizing antibodies were detected in 47 of 68 (69%) patients who were seropositive at baseline (after primary vaccination). The correlation between seropositivity and presence of neutralizing antibodies was statistically significant (p<0.001, Fisher's exact test).

At 4 weeks post-third dose, samples from 100 patients were available for testing. Eighty-five of these patients were seropositive at 4 weeks and 15 were seronegative. Neutralizing antibodies were detected in 77 of 85 (91%) seropositive patients at 4 weeks. The correlation between seropositivity and presence of neutralizing antibodies was also statistically significant at 4 weeks (p<0.001, Fisher's exact test).

We also analyzed the correlation of anti-S titers at baseline and 4 weeks to the percentage of virus neutralization, with 30% or more neutralization being consistent with positive result for detection of neutralizing antibodies. We observed that at baseline and 4 weeks, anti-S titers correlated with percentage of viral neutralization with higher titers correlating with higher percentage of viral neutralization (*Figure 1H* <0.001 by Spearman rank correlation). These results are summarized in *Table 2*.

### Neutralization against Omicron BA.1

Thirty-five patients were found be seronegative after the third dose. Due to the emergence of the Omicron BA.1 wave, we further assessed neutralization activity for the seronegative cohort (N=35)

**Table 2.** Results for third dose of vaccine.

| Spike antibody results | n=106 | | | |
|---|---|---|---|---|
| | Four-week negative | Four-week positive | Seroconversion rate | p value |
| Baseline negative | 15 | 20 | 57% | <0.001* |
| Baseline positive | 0 | 71 | | |
| Total | 15 | 91 | | |
| | | | | |
| Rise in spike antibody titers overall (AU/mL) | Median | IQR | | |
| Titer at baseline | 212.1 | 50–2873 | | |
| Titer at 4 weeks | 9997 | 880.7–47,063 | | |
| | | | | |
| Rise in spike antibody titers (AU/mL) | Median | IQR | | |
| Hematologic malignancy | 2167 | 0–10,131 | | <0.001* |
| Solid malignancy | 31,010 | 9531–44,464 | | |
| Rise in spike antibody titers by solid/lymphoid/myeloid (AU/mL) | | | | |
| Lymphoid cancers | 1169 | 0–8661 | | <0.001* |
| Myeloid cancers | 9424 | 4381–20,444 | | |
| Solid cancers | 31,010 | 9531–44,464 | | |
| | | | | |
| Association with certain cancer-directed therapies | | | | |
| Bruton's tyrosine kinase inhibitors | | | | |
| Change in spike antibody titers (AU/mL) | Median | IQR | | |
| Patients on BTKi (n=12) | 0 | 0–3393 | | <0.001* |
| Patients not on BTKi | 9355 | 877.3–34,410 | | |
| Anti-CD20 antibody treatment | | | | |
| Change in spike antibody titers (AU/mL) | Median | IQR | | |
| Patients on CD20 (n=25) | 0 | 0–910.5 | | 0.0133* |
| Patients not on CD20 | 12735 | 2842–38,863 | | |
| Anti-CD20 antibody treatment within 6 months | Median | IQR | | |
| Yes | 0 | 0–0 | | 0.05482 |
| No | 587 | 0–4314 | | |
| Change in spike antibody titer by prior COVID infection | Median | IQR | | |

*Table 2 continued on next page*

*Table 2 continued*

| Spike antibody results | | n=106 | | |
| --- | --- | --- | --- | --- |
| Yes (n=9) | 19,350 | 9286–32,151 | | 0.3051 |
| No (n=96) | 6706 | 444.1–33,831 | | |
| Change in spike antibody titer by type of booster given | Median | IQR | | |
| BNT162b2 | 5534 | 433.8–18,074 | | 0.09014 |
| mRNA-1273 | 31451 | 515.5–45,057 | | |
| Change in spike antibody titer by age | Median | IQR | | |
| Age <65 years | 27451 | 2641–50,000 | | 0.03438* |
| Age ≥65 years | 6152 | 558.9–41,765 | | |
| T-cell activity | | | | |
| Baseline | n=88 | % | | |
| Positive | 65 | 74% | | |
| Negative | 23 | 26% | | |
| Four-week | n=89 | | | |
| Positive | 76 | 85% | | |
| Negative | 13 | 15% | | |

Baseline neutralization activity assay (all evaluable patients, WT virus)

| | Anti-S antibody negative | Anti-S antibody positive | Total | p value |
| --- | --- | --- | --- | --- |
| Neutralizing antibodies detected | 0 | 47 | 47 | <0.001 |
| Neutralizing antibodies not detected | 35 | 21 | 56 | |
| Total | 35 | 68 | 103 | |

Four-week neutralization activity assay (all evaluable patients, WT virus)

| | Anti-S antibody negative | Anti-S antibody positive | Total | p value |
| --- | --- | --- | --- | --- |
| Neutralizing antibodies detected | 0 | 77 | 77 | <0.001 |
| Neutralizing antibodies not detected | 15 | 8 | 23 | |
| Total | 15 | 85 | 100 | |

Four-week neutralization assay (seronegative cohort 4 weeks) n=35

*Table 2 continued*

| Spike antibody results | | n=106 |
| --- | --- | --- |
| Wild type | | |
| Negative | 19 | 54% |
| Positive | 16 | 46% |
| | | |
| Omicron | | |
| Negative | 29 | 83% |
| Positive | 6 | 17% |

*Statistically significant.

against WT SARS-CoV-2 and BA1.1.529 (Omicron BA.1). At 4 weeks (after third dose) neutralization was noted in 46% patients (16/35) for the WT virus while only 17% of patients had detectable neutralization activity (6/35) for the Omicron variant. These results are summarized in *Table 2*.

## Six-month follow-up post-third dose of vaccine

Forty-seven patients (44%) out of 106 completed 4–6 months' follow-up for the third dose study. All these patients were seropositive 4 weeks after the third dose and strikingly, we observed that all patients maintained a positive anti-S antibody at 4–6 months' follow-up. Eleven of these 47 patients had solid malignancies and 36 had hematologic malignancies. Six patients had received anti-COVID monoclonal antibody (moAb) therapy as per standard of care (4 tixagevimab-cilgavimab or Evusheld, 1 casirivimab/imdevimab or regen-co-v, and 1 sotrovimab between the 4 week and 4–6 months' follow-up). A striking increase in titers in this small cohort of patients was noted to a median titer of 17481.2 AU/mL. Four patients had breakthrough SARS-CoV-2 infections and 9 patients had received a fourth dose of COVID-19 vaccine outside of the context of the study prior to the time of 4–6 months' follow-up. Of the four breakthrough infections, one patient had no symptoms and three had mild symptoms. The median decline in titer for 41 patients who did not receive anti-SARS-CoV-2 (anti-S) moAb treatment in the interim to confound results was –922.2 AU/mL. When compared to the antibody levels 4 weeks after booster vaccination, the median percentage decline in titers was 56.4%. However, despite the noted decline not a single patient in this cohort seroreverted (*Figure 1I*), especially when compared to decline post-two vaccines. In our initial report of seroconversion post-third vaccine, we reported waning of immunity in 99 patients post-two vaccines. The median decline in the 99 patient cohort was 72.1% with two patients losing detectable antibody response (*Shapiro et al., 2022*).

## Efficacy of fourth dose vaccine for patients that were seronegative or low seropositive after third dose

### Baseline characteristics

Eighteen patients were enrolled into the fourth dose study. Median age for this cohort was 69.5 years (IQR 65.5–73.8). Thirty-nine percent (7/18) were seronegative at baseline (after three

**Table 3.** Baseline characteristics of the fourth dose cohort.

| | N (%) |
| --- | --- |
| Baseline seronegative | 7 (39%) |
| Baseline low positive (spike ab <1000 AU/mL) | 11 (61%) |
| Cancer diagnosis | |
| CLL | 7 (39%) |
| Waldenstrom's macroglobulinemia | 3 (17%) |
| DLBCL | 2 (11%) |
| Multiple myeloma | 2 (11%) |
| Mantle cell Lymphoma | 1 (6%) |
| Marginal zone lymphoma | 1 (6%) |
| Hodgkins lymphoma | 1 (6%) |
| MDS | 1 (6%) |
| Fourth dose vaccine type | |
| BNT162b2 | 15 (83%) |
| Ad26.CoV2.S | 3 (17%) |

Table 4. Correlation of fourth dose vaccine response with baseline characteristics.

| | Non-responder (n=6) | Responder (n=12) | p value |
|---|---|---|---|
| Age | 79.5 | 67.5 | 0.01293* |
| Baseline WBC | 4.95 | 5.15 | 0.45 |
| Baseline ANC | 2.6 | 3.5 | 0.26 |
| Baseline ALC | 1.2 | 1.3 | 0.57 |
| Baseline AMC | 0.5 | 0.65 | 0.73 |
| Baseline absolute CD3 | 773 | 835.5 | 0.57 |
| Baseline absolute CD4 | 406.5 | 407.5 | 0.71 |
| Baseline absolute CD8 | 310 | 247 | 0.40 |
| Baseline absolute CD19 | 1 | 113.5 | 0.04874* |
| Baseline absolute CD16/56 | 243.5 | 200 | 0.57 |
| Baseline IgG | 777 | 757 | 0.51 |
| Baseline IgA | 90.5 | 118 | 0.57 |
| Baseline IgM | 17 | 60.5 | 0.001442[†] |
| 4-Week WBC | 5.1 | 5.8 | 0.40 |
| 4-Week ANC | 2.7 | 3.45 | 0.57 |
| 4-Week ALC | 1.1 | 1.4 | 0.60 |
| 4-Week AMC | 0.55 | 0.65 | 0.60 |
| 4-Week absolute CD3 | 754 | 983 | 0.40 |
| 4-Week absolute CD4 | 461.5 | 369.5 | 0.93 |
| 4-Week absolute CD8 | 297.5 | 269 | 0.40 |
| 4-Week absolute CD19 | 2.5 | 105 | 0.07 |
| 4-Week absolute CD16/56 | 232.5 | 219 | 0.93 |
| 4-Week IgG[†] | 741.5 | 832 | 0.62 |
| 4-Week IgA[†] | 86 | 112 | 0.69 |
| 4-Week IgM[†] | 15 | 62 | 0.003561[†] |

*Statistically significant.
[†]n=11.

doses, pre-fourth dose) and 61% patients (11/18) were sero-low (anti-S ab <1000 AU/mL). All patients had hematologic malignancies in this cohort and the breakdown of diagnoses is provided in *Table 3*. Eighty-three percent of the patients (15/18) received BNT162b2 fourth booster shots and 17% (3/18) patients received Ad26.CoV2.S as their fourth booster vaccine. In addition, we also measured CBC, lymphocyte subsets, immunoglobulin G, A, and M (quantitative Ig) levels at baseline (pre-fourth dose) and 4 weeks (post-fourth dose) (*Table 4*). The median time between second and third vaccination was 167 days (5.5 months) and that between third and fourth vaccination was 155 days (5.1 months).

## Anti-spike IgG responses after the fourth dose

A patient was classified as a responder if they (1) had positive anti-S antibody at 4 weeks if seroneg-ative at baseline (after three doses) or (2) if they achieved a titer of >1000 AU/mL at 4 weeks if they were sero-low at baseline (after three doses). As such, we observed a 67% response rate (12/18) in patients for the fourth dose cohort. Two of seven seronegative patients seroconverted to positive anti-S antibody at 4 weeks with a seroconversion rate of 29% in this cohort. All sero-low patients (11/11) responded with an IgG level >1000 after the fourth dose (*Figure 2A*). For the whole cohort,

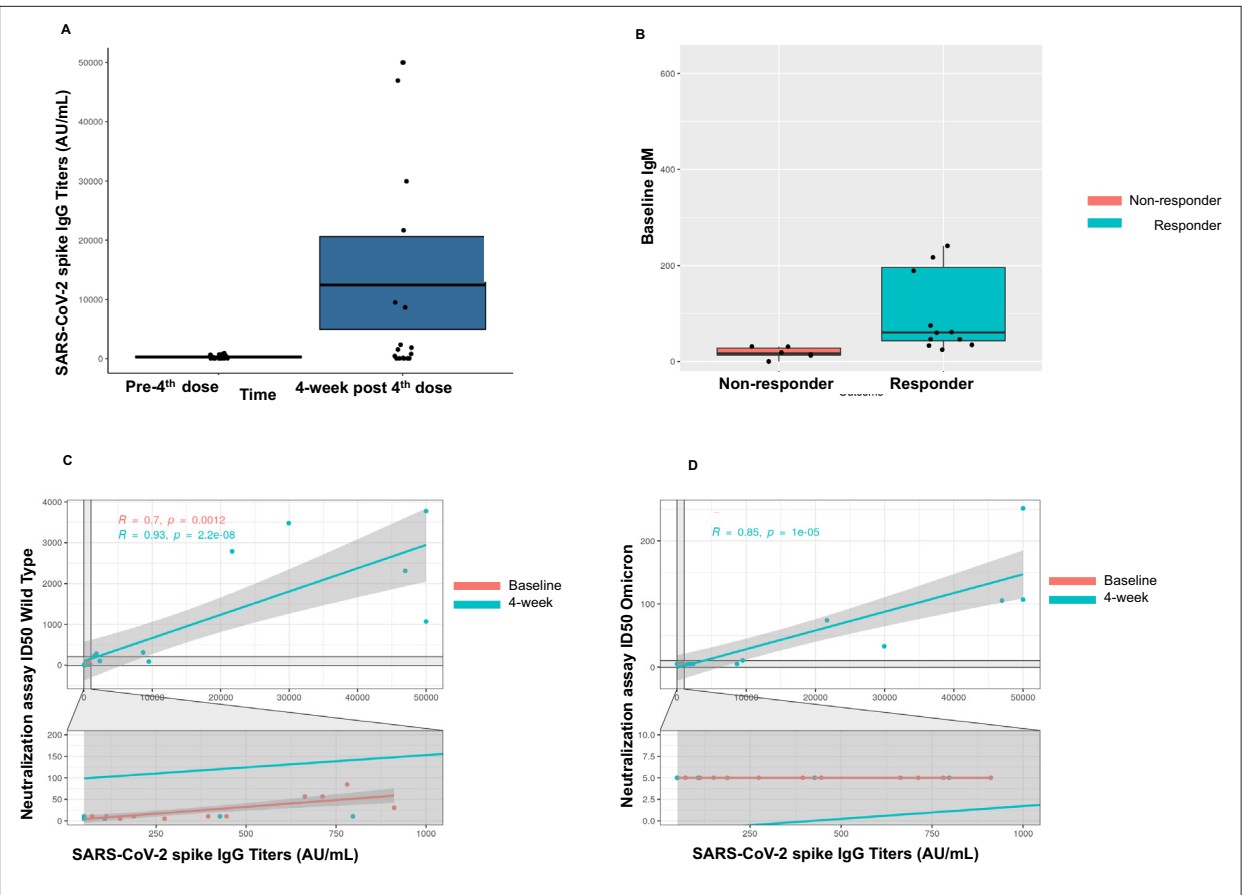

**Figure 2.** Immunogenicity of the fourth dose of coronavirus disease 2019 (COVID-19) vaccine in cancer patients with seronegativity after three doses. (**A**) Anti-spike IgG levels after the fourth dose of COVID-19 vaccine for the entire cohort n=18. (**B**) Correlation of baseline IgM levels with response to fourth dose of vaccine, n=18 Kruskal-Wallis test. (**C**) Line diagram showing correlation between anti-SARS-CoV-2 (anti-S) titer and neutralization activity for wild-type (WT) virus at baseline and 4 weeks, n=18, Spearman's test. (**D**) Line diagram showing correlation between titer and neutralization activity for Omicron strain at baseline and 4 weeks n=18, Spearman's test. All statistical tests performed at a pre-determined threshold of p<0.05 for statistical significance.

the median anti-S antibody at baseline (after three doses) was 131.1 AU/mL (<50–432.9 AU/mL) and at 4 weeks (after fourth dose) was 1700 AU/mL (IQR 64.3–18,627 AU/mL). The two patients that seroconverted after fourth dose both had a diagnosis of Waldenstrom's macroglobulinemia. Both patients had received anti-CD20 antibody and chemotherapy as part of their treatment. One patient was actively on a BTK inhibitor and the second patient was off active treatment at the time of study participation. We further investigated association of baseline laboratory values, such as CBC, lymphocyte subsets, and quantitative Ig levels and observed that patients in the responder group had higher baseline IgM (60.5 mg/dL) compared with the non-responder group (median 17 mg/dL, p<0.001) (*Figure 2B*). Additionally, we also observed that the median CD19+ cell count was significantly lower in the non-responder group versus the responder group (1 vs. 113, p=0.04). No patients were on intravenous immunoglobulin at the time of study participation. These results are summarized in *Table 5*.

### T-cell activity against SARS-CoV-2 after the fourth dose

T-cell activity was assessed at baseline (pre-fourth dose) and at 4 weeks (post-fourth dose) using the SARS-CoV-2 IGRA. At baseline, 14 patients had evaluable T-cell responses and a positive response was noted in 79% patients (11/14). Of these, three patients had negative anti-S antibody at baseline. At 4 weeks after the fourth dose, a positive T-cell response was seen in 17/18 (94%) patients. These results are summarized in *Table 5*.

**Table 5.** Results for fourth dose study.

| Overall response | 18 | | |
|---|---|---|---|
| Responder | 12 | 67% | |
| Non-responder | 6 | 33% | |
| Median age | | IQR | |
| Responder | 67.5 | 63.75–70.75 | 0.01293* |
| Non-responder | 79.5 | 72.75–81.75 | |
| Median baseline IgM | | | |
| Responder | 60.5 | | 0.001442 * |
| Non-responder | 17 | | |
| Median spike antibody at baseline (AU/mL) | 131.1 | <50–432.9 | |
| Median spike antibody at 4 weeks (AU/mL) | 1700 | 64.3–18627 | |
| T-cell activity at baseline | n=14 | | |
| Positive | 11 | 79% | |
| Negative | 3 | 21% | |
| T-cell activity at 4 weeks | n=18 | | |
| Positive | 17 | 94% | |
| Negative | 1 | 6% | |
| Baseline | | | |
| Neutralization assay baseline | Negative | Positive | |
| WT | 6 (33%) | 12 (67%) | |
| Omicron | 18 (100%) | 0 (0%) | |
| | | | |
| Neutralization assay 4 week | Negative | Positive | |
| WT | 5 (28%) | 13 (72%) | |
| Omicron | 12 (67%) | 6 (33%) | |

*Statistically significant.

## Neutralization activity against SARS-CoV-2 after fourth dose

We also assessed neutralization activity at baseline (pre-fourth dose) and at 4 weeks (post-fourth dose) against WT and Omicron (B.1.1.529, BA.1). The results are summarized in *Table 5*. Overall, neutralization activity was seen in 67% patient samples at baseline and in 72% patient samples at 4 weeks. Strikingly, neutralization activity against Omicron was absent in all patient samples at baseline, however became detectable in 33% (6/18) patients at 4 weeks after the fourth dose. The titer of anti-S antibody correlated with neutralization activity at baseline and at 4 weeks against the WT virus (p<0.001) (*Figure 2C*). We also observed correlation between the titer of anti-S antibody with neutralization activity at 4 weeks for the Omicron variant (*Figure 2D*).

## Exploratory analysis for immunoglobulin levels

The observation for baseline IgM correlating with response to the fourth dose of the COVID-19 vaccine led us to perform an exploratory analysis to assess if IgG and IgA levels would also correlate with the response. Given that our fourth dose cohort was small, we performed this exploratory analysis by combining the baseline immunoglobulin levels for the baseline seronegative cohort for the third dose study (n=35) and baseline immunoglobulin levels for the fourth dose study (n=18). In this exploratory analysis, we observed that the median levels for all immunoglobulin subtypes were lower in patients

who either did not seroconvert after the third dose or did not respond to the fourth dose (IgA 49 mg/dL vs. 116.5 mg/dL, IgM 16.6 mg/dL vs. 48.3 mg/dL, IgG 488 mg/dL vs. 759.5 mg/dL with p values of 0.05, 0.002, and 0.006, respectively [Kruskal Wallis test]).

## Discussion

Since the authorization of third doses for patients with a weakened immune system, several studies have shown enhanced immunogenicity for a third dose of COVID-19 vaccine in patients with cancer (*Shapiro et al., 2022*; *Munro et al., 2021*; *Shroff et al., 2021*). In particular, patients with lymphoid malignancies have been consistently shown to have reduced seroconversion after two doses of the COVID-19 vaccines (*Greenberger et al., 2021*; *Perry et al., 2021*; *Herishanu et al., 2021*; *Ghione et al., 2021*). Studies looking at immunogenicity of a third dose of COVID-19 vaccines have reported that a subset of these patients can be induced to have an immune response with the third dose of the COVID-19 vaccines (*Shapiro et al., 2022*; *Lim et al., 2022*).

Correlation between anti-S antibody titers and neutralization activity in patients with cancer has been demonstrated (*Mack et al., 2022*). However, with the emergence of the Omicron (B.1.1.529) variant which was discovered in November 2021 and then spread quickly globally, the situation changed. Omicron, with its extensive mutations in neutralizing epitopes, is able to at least partially evade in vitro neutralizing antibodies induced by third doses in patients with cancer (*Mack et al., 2022*; *Chang et al., 2022*). The potential utility and timing of a fourth COVID-19 vaccine dose has been brought up especially for those who are at risk for poor seroconversion after third doses (*Ehmsen et al., 2022*), with the CDC recommending two additional boosters following a three-vaccine primary series (*Centers for Disease control and prevention, 2023*). These variants in part overcome vaccine-induced immunity and are resistant to many of the available monoclonal antibody products (*Mack et al., 2022*; *Chang et al., 2022*; *Zhou et al., 2022*).

Our results demonstrate that a third dose of COVID-19 vaccine boosts detectable anti-S immunity in the majority of cancer patients and can seroconvert a subset of them not responding to primary two-vaccine doses. The third COVID-19 vaccine also results in boosting of T-cell responses and leads to a rise in neutralizing antibodies. Patients who have received anti-CD20 antibody therapy or BTK inhibitors remain at risk for lower seroconversion whereas those who have been infected with COVID-19 in the past have a very strong immune response likely due to immunologic memory. Our results show that the higher the titer of the anti-S antibody, the higher likelihood of neutralization in a surrogate neutralization assay adding to the evidence that this may be a good strategy to prevent symptomatic SARS-CoV-2 infection as well as an appropriate surrogate marker to guide research and clinical management (*Khoury et al., 2021*). Our study also provides the reassuring finding that the large majority of patients with cancer retain detectable humoral immunity at 6 months' post-third dose of COVID-19 vaccination. While we do not have an internal control group of non-cancer patients, previous studies have reported a similar boosting of immune responses in the general population after third dose, waning of immunity, and another boost of immune response after fourth dose (*Goldberg et al., 2022*; *Cohen et al., 2022*).

Reports of efficacy of fourth COVID-19 vaccine doses are emerging. A study from Israel demonstrated enhanced Omicron neutralization after a fourth dose of COVID-19 mRNA vaccine in healthy healthcare workers (*Regev-Yochay et al., 2022*). However a study of 25 patients with solid organ transplant recipients showed that the fourth dose was not effective in inducing Omicron neutralization (*Karaba et al., 2022*). Such a study has not been published yet for patients with cancer, making this an unmet need. We designed a prospective cohort study of a fourth dose of the COVID-19 vaccine in patients with cancer precisely to address this question. Our results suggest that in cohorts of highly immune suppressed patients, especially those on B-cell depleting treatments such as anti-CD20 antibodies and BTK inhibitors, a baseline assessment of immunity based on prior treatment history and immunological markers such as IgM levels and CD19+ cell levels may help predict the response to COVID-19 vaccinations and support administration of additional vaccine doses. Notably, serum IgM levels were previously shown to correlate with mRNA vaccine responses of solid organ transplant recipients (*Azzi et al., 2021*). In addition, further testing to assess serological and cellular markers of the response may be helpful to identify the patients at highest risk to prioritize these patients for preventive/prophylactic strategies as well as enrichment markers for further experimental studies. Finally, the fourth vaccine dose results in a significant increase in anti-spike antibodies in low seropositive

patients and seroconversion in a proportion of seronegative immunosuppressed patients with cancer. However, caution should be exercised in generalizing these results to the broader immunosuppressed population given the small sample size of our cohort and the disproportionately high representation of hematologic malignancy patients. Similar to previous reports, the additional doses do lead to enhanced neutralization activity against the WT virus, but not the Omicron (BA.1) variant. Future efforts are needed to evaluate variant-specific vaccines as well as additional protective measures, such as passive immunization strategies, especially for this immunosuppressed patient population that may not benefit as much as healthy controls from booster doses of existing vaccines. The bivalent COVID-19 vaccine was introduced after the enrollment for our study was closed, however it is reassuring to see that the bivalent vaccine has better neutralization activity against Omicron sub-variants (*Davis-Gardner et al., 2023*). Ongoing monitoring of variants and the proposal for annual vaccination by the FDA are important next steps that will be crucial in keeping the circulating SARS-CoV-2 levels at reasonable levels (*Scribd, 2023*). Further efforts are also needed to better determine cutoff values at which anti-S antibody levels provide protection from symptomatic COVID-19. At the present time, this data exists only for neutralizing antibody titers (*Khoury et al., 2021*; *Gilbert et al., 2022*) and the commercially available anti-S antibody assays are quite heterogenous with efforts being made to improve equivalency in titer reporting (*Infantino et al., 2021*). Our study while providing a correlation between anti-S antibody titer and neutralizing antibody titer supports that the higher the titer, the better neutralization is expected and by extrapolation, less likelihood of symptomatic infection, however this needs to be confirmed in larger, systematic studies.

## Acknowledgements

This study was supported with funding from the National Cancer Institute Grant 3P30CA013330-49S3 and NCORP Grant 2UG1CA189859-06. The authors also acknowledge support from the Jane and Myles Dempsey Family and Leukemia Lymphoma Society (grant # IRVBC0004-22). The funders had no role in study design, data collection and analysis, decision to publish, or preparation of the manuscript. Work in the Krammer laboratory was partially funded by the Centers of Excellence for Influenza Research and Surveillance (CEIRS, contract # HHSN272201400008C), the Centers of Excellence for Influenza Research and Response (CEIRR, contract # 75N93021C00014), by the Collaborative Influenza Vaccine Innovation Centers (CIVICs contract # 75N93019C00051) and by institutional funds. Finally, this effort was also supported by the Serological Sciences Network (SeroNet) in part with Federal funds from the National Cancer Institute, National Institutes of Health, under Contract No. 75N91019D00024, Task Order No. 75N91020F00003. The content of this publication does not necessarily reflect the views or policies of the Department of Health and Human Services, nor does mention of trade names, commercial products or organizations imply endorsement by the U.S. Government.

## Additional information

### Competing interests

Maite Sabalza: is affiliated with EUROIMMUN and has no financial interests to declare. Florian Krammer: The Icahn School of Medicine at Mount Sinai has filed patent applications relating to SARS-CoV-2 serological assays and NDV-based SARS-CoV-2 vaccines which list Florian Krammer as co-inventor. Mount Sinai has spun out a company, Kantaro, to market serological tests for SARS-CoV-2. Florian Krammer has consulted for Merck and Pfizer (before 2020), and is currently consulting for Pfizer, Seqirus, 3rd Rock Ventures and Avimex. The Krammer laboratory is also collaborating with Pfizer on animal models of SARS-CoV-2.(Serological Assay, U.S. Application Serial No. 17/913,783, NDV-HXP-S, U.S. Application Serial No. 17/922,777). Amit Verma: Reviewing editor, eLife. The other authors declare that no competing interests exist.

### Funding

| Funder | Grant reference number | Author |
|---|---|---|
| National Cancer Institute | 3P30CA013330-49S3 | Balazs Halmos |

| Funder | Grant reference number | Author |
| --- | --- | --- |
| National Cancer Institute | Community Oncology Outreach Program 2UG1CA189859-06 | Balazs Halmos |
| Leukemia & Lymphoma Society | IRVBC0004-22 | Amit Verma Balazs Halmos |
| Centers of Excellence for Influenza Research and Surveillance | HHSN272201400008C | Florian Krammer |
| Centers of Excellence for Influenza Research and Response | 75N93021C00014 | Florian Krammer |
| Collaborative Influenza Vaccine Innovation Centers | 75N93019C00051 | Florian Krammer |
| National Cancer Institute | 75N91019D00024 | Florian Krammer |

The funders had no role in study design, data collection and interpretation, or the decision to submit the work for publication.

## Author contributions

Astha Thakkar, Conceptualization, Data curation, Methodology, Writing – original draft, Writing – review and editing; Kith Pradhan, Software, Formal analysis, Writing – review and editing; Benjamin Duva, Data curation, Writing – original draft, Project administration, Writing – review and editing; Juan Manuel Carreno, Srabani Sahu, Victor Thiruthuvanathan, Sonia Gallego, Tushar D Bhagat, Johanna Rivera, Gaurav Choudhary, Raul Olea, Maite Sabalza, Investigation, Methodology, Writing – review and editing; Sean Campbell, Investigation, Methodology, Project administration, Writing – review and editing; Lauren C Shapiro, Conceptualization, Data curation, Writing – review and editing; Matthew Lee, Ryann Quinn, Ioannis Mantzaris, Data curation, Writing – review and editing; Edward Chu, Resources, Supervision, Project administration, Writing – review and editing; Britta Will, Supervision, Investigation, Methodology; Liise-anne Pirofski, Conceptualization, Supervision, Writing – review and editing; Florian Krammer, Supervision, Investigation, Methodology, Writing – review and editing; Amit Verma, Conceptualization, Supervision, Funding acquisition, Writing – original draft, Writing – review and editing; Balazs Halmos, Conceptualization, Resources, Data curation, Supervision, Funding acquisition, Methodology, Project administration, Writing – review and editing

## Author ORCIDs

Astha Thakkar  http://orcid.org/0000-0001-7778-0752
Gaurav Choudhary  http://orcid.org/0000-0001-5365-6706
Amit Verma  http://orcid.org/0000-0002-5408-1673
Balazs Halmos  http://orcid.org/0000-0001-7548-8360

## Ethics

Clinical trial registration NCT05016622.
The study was approved by Montefiore-Einstein institutional review board (IRB# 2021-13204). Participants were recruited after referral to the study from their treating oncologists. At the consent visit, patients were provided with a study overview including initial lab draw, vaccine appointment and follow-up at pre-specified time points. The informed consent document included consent for research samples and consent to protect confidential patient information by the personnel approved under the IRB. Any person not involved with the research study did not have access to patient identifying data. De-identified data was allowed to be shared with collaborators and findings from the study be published. The informed consent document also included consent for a future research lab draw should an improved test for SARS-CoV-2 immunity became available. Finally, the consent included patient's right to withdraw from the study at any time. The patient was provided with a copy of the signed informed consent.

## Decision letter and Author response

Decision letter https://doi.org/10.7554/eLife.83694.sa1
Author response https://doi.org/10.7554/eLife.83694.sa2

## Additional files

### Supplementary files
• MDAR checklist

### Data availability
The data for this clinical trial contains protected health information for the participants that includes a large amount of information as to specific dates/treatments/cancer diagnoses. Therefore, the the de-identified dataset is available upon request to allow researchers access to complete data while protecting potentially identifiable patient-level information. Future research using the dataset will need to be done in accordance with standardized guidelines and with local ethics approval. Computer code has been deposited in GitHub and can be found at https://github.com/kith-pradhan/Covid-Booster (copy archived at *Thakkar, 2023a*) and https://github.com/kith-pradhan/CovidBooster4th (copy archived at *Thakkar, 2023b*).

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
