## [Editor Report]

This important study evaluates the immunogenicity of 3rd and 4th doses of SARS-CoV2 vaccinations in patients with cancer. Their study is notable in that neutralization of Omicron was absent in all patients after the third dose but increased to 33% after the fourth dose. With the definitions and patient population better described, this paper would be of interest to those studying the effects of repeated COVID boosters on Omicron immunity.

---

## [Decision Letter]

**Decision letter after peer review:**

Thank you for submitting your article "Efficacy and longevity of immune response to COVID-19 vaccine boosters in severely immunocompromised patients with cancer: a single arm clinical trial" for consideration by *eLife*. Your article has been reviewed by 2 peer reviewers, and the evaluation has been overseen by a Reviewing Editor and Mone Zaidi as the Senior Editor. The reviewers have opted to remain anonymous.

Essential revisions:

Overall the reviewers were very positive. However, two themes emerged:

1) Clarification of the wording and definitions were highlighted by both reviewers. Please pay close attention to clearly defining each dose and each patient condition as the reviewers note this can be confusing.

2) Please address confounding factors for the selected patient populations.

*Reviewer #1 (Recommendations for the authors):*

This is a timely study in a diverse cohort of immunocompromised cancer patients. The correlation of doses, therapies with anti-S antibodies as well as assessment of cellular (T cell) immune responses and viral neutralization against wild-type as well as Omicron variant is commendable.

*Reviewer #2 (Recommendations for the authors):*

1. The study population and timing of booster vaccinations are unclear from the title and abstract. Specifically, what constitutes "severely immunocompromised patients?" While patients with hematological malignancies, particularly those receiving B-cell-depleting therapies, have been susceptible to COVID-19, approximately 40% of the patients in this study had solid tumors. What makes them "severely immunocompromised?" Furthermore, the abstract conclusion would give the reader the impression that this study was only performed on hematologic malignancy patients.

2. Ultimately, measurements of anti-spike antibodies and neutralization are surrogates for immunity to SARS-CoV-2. Presenting the data as binary data (e.g., responder versus non-responder; seroconversion versus no seroconversion) implies that there are clear cutoffs to define protection. Likewise, in the abstract, mention is made of an "adequate immune boost." How is this defined?

3. Admittedly, the nomenclature surrounding additional doses of COVID-19 vaccines has been very confusing in the lay press and medical community. Nonetheless, the nomenclature used in the manuscript is also confusing at times. For example, in the introduction, the authors mention "a 3rd booster dose." What does this mean? A third overall dose or a third "booster" dose (i.e., a 5th total dose) on top of the original series? Likewise, the authors frequently present "baseline data" but it's unclear to what baseline they are referring (i.e., pre-3rd dose, pre-4th dose, etc).

4. Methods: page 5, line 112: "We previously reported preliminary findings of a 56% seroconversion rate for patients with cancer who did not have a detectable immune response after 2 doses." This statement is unclear. Presumably, this is after a third dose. And was this based upon the "3rd dose study" described immediately above?

5. Page 10, lines 301-302: Of the 21 patients who were seronegative for anti-S antibody at baseline, how many had T cell responses?

---

## [Author Response]

Essential revisions:Overall the reviewers were very positive. However, two themes emerged:1) Clarification of the wording and definitions were highlighted by both reviewers. Please pay close attention to clearly defining each dose and each patient condition as the reviewers note this can be confusing.

We have now added text in parenthesis clarifying at each instance the dose of the vaccine we referred to.

2) Please address confounding factors for the selected patient populations.

We have added text in results for the comment regarding IVIG (lines 393-394):

“No patients were on intravenous immunoglobulin (IVIG) at the time of study participation”

Reviewer #2 (Recommendations for the authors):1. The study population and timing of booster vaccinations are unclear from the title and abstract. Specifically, what constitutes "severely immunocompromised patients?" While patients with hematological malignancies, particularly those receiving B-cell-depleting therapies, have been susceptible to COVID-19, approximately 40% of the patients in this study had solid tumors. What makes them "severely immunocompromised?" Furthermore, the abstract conclusion would give the reader the impression that this study was only performed on hematologic malignancy patients.

Thank you for the insightful comments! We have added text in the discussion to further clarify “severely immunocompromised” (lines 464-465) and modified the abstract for clarity as well.

“especially those on B-cell depleting treatments such as anti-CD20 antibodies and BTK inhibitors,”

2. Ultimately, measurements of anti-spike antibodies and neutralization are surrogates for immunity to SARS-CoV-2. Presenting the data as binary data (e.g., responder versus non-responder; seroconversion versus no seroconversion) implies that there are clear cutoffs to define protection. Likewise, in the abstract, mention is made of an "adequate immune boost." How is this defined?

We have added text to the discussion part expanding on the above comment (483-494)

“Further efforts are also needed to better determine cut-off values at which anti-S antibody levels provide protection from symptomatic COVID-19. At the present time, this data exists only for neutralizing antibody titers[36, 44] and the commercially available anti-S antibody assays are quite heterogenous with efforts being made to improve equivalency in titer reporting[45]. Our study while providing a correlation between anti-S antibody titer and neutralizing antibody titer supports that the higher the titer, the better neutralization is expected and by extrapolation, less likelihood of symptomatic infection however this needs to be confirmed in larger, systematic studies.”

3. Admittedly, the nomenclature surrounding additional doses of COVID-19 vaccines has been very confusing in the lay press and medical community. Nonetheless, the nomenclature used in the manuscript is also confusing at times. For example, in the introduction, the authors mention "a 3rd booster dose." What does this mean? A third overall dose or a third "booster" dose (i.e., a 5th total dose) on top of the original series? Likewise, the authors frequently present "baseline data" but it's unclear to what baseline they are referring (i.e., pre-3rd dose, pre-4th dose, etc).

We have clarified the language throughout the manuscript

4. Methods: page 5, line 112: "We previously reported preliminary findings of a 56% seroconversion rate for patients with cancer who did not have a detectable immune response after 2 doses." This statement is unclear. Presumably, this is after a third dose. And was this based upon the "3rd dose study" described immediately above?

The text in methods for 3^rd^ dose study has now been modified to clarify the number and timing of vaccine doses.

5. Page 10, lines 301-302: Of the 21 patients who were seronegative for anti-S antibody at baseline, how many had T cell responses?

We have clarified that the 21 patients are a subset of 65 patients that had positive T-cell responses, therefore all of them had positive T-cell response (line 311-312).